# Characterization of Regulatory T Cells in Patients Infected by *Leishmania Infantum*

**DOI:** 10.3390/tropicalmed8010018

**Published:** 2022-12-27

**Authors:** Rephany F. Peixoto, Bruna M. Gois, Marineuma Martins, Pedro Henrique S. Palmeira, Juliana C. Rocha, Juliana A. S. Gomes, Fátima L. A. A. Azevedo, Robson C. Veras, Isac A. de Medeiros, Teresa C. S. L. Grisi, Demétrius A. M. de Araújo, Ian P. G. Amaral, Tatjana S. L. Keesen

**Affiliations:** 1Immunology of Infectious Diseases Laboratory, Department of Cellular and Molecular Biology, Federal University of Paraiba, João Pessoa 58051-900, Brazil; 2Department of Morphology, Institute of Biological Sciences, Federal University of Minas Gerais, Pampulha, Belo Horizonte 31270-901, Brazil; 3Research Institute for Drugs and Medicines, Federal University of Paraíba, João Pessoa 58051-900, Brazil; 4Department of Biotechnology, Federal University of Paraiba, João Pessoa 58051-900, Brazil

**Keywords:** regulatory T cells, human leishmaniasis, VL immunomodulation

## Abstract

High IL-10 levels are pivotal to parasite survival in visceral leishmaniasis (VL). Antigenic stimuli induce IL-10 expression and release of adenosine by CD39/CD73. Due their intrinsic ability to express IL-10 and produce adenosine from extracellular ATP, we evaluated the IL-10, CD39, and CD73 expression by Regulatory T cells (Treg) correlated with VL pathology. Using flow cytometry, Treg cells was analyzed in peripheral blood samples from VL patients (in the presence and absence of *Leishmania infantum* soluble antigen (SLA)) and healthy individuals (negative endemic control—NEC group), without any treatment. Additionally, IL-10 levels in leukocytes culture supernatant were measured in all groups by ELISA assay. VL patients presented more Treg frequency than NEC group, independently of stimulation. ELISA results demonstrated that SLA induced higher IL-10 expression in the VL group. However, the NEC group had a higher Treg IL-10^+^ compared to the VL group without stimulation and SLA restored the IL-10 in Treg. Additionally, an increase in Treg CD73^+^ in the VL group independently of stimuli compared to that in the NEC group was observed. We suggest that Treg are not the main source of IL-10, while the CD73 pathway may be an attempt to modulate the exacerbation of immune response in VL disease.

## 1. Introduction

*Leishmania* sp. is an intracellular pathogen that mainly infects phagocytes, causing leishmaniasis, an infectious anthropozoonotic disease reported as the second leading cause of death from parasitic infections worldwide. Leishmania infantum causes the most severe clinical form of the disease called visceral leishmaniasis (VL). Main clinical symptoms are fever, hepatosplenomegaly, and pancytopenia [1,2]. 

Pathogenesis of VL is characterized by the generation of a favorable microenvironment for the survival of protozoa and disease progression, mainly due to the suppression of a Th1 profile response and activation of IFN-y-responsive macrophages mediated by the high production of regulatory cytokines, particularly IL-10 [3,4,5]. IL-10 plays a central role in the course of the immune response present in VL, contributing to the chronicity of the disease through mechanisms that compromise the migration and activation of leukocytes mediated by metabolites such as prostaglandins E and J and adenosine [6,7,8,9]. Adenosine, specifically produced through the hydrolysis of extracellular ATP by ectonucleotidases CD39/CD73, also contributes to the attenuation of the inflammatory response in leishmaniasis. It mainly reduces pro-inflammatory cytokine production, down-regulates of nitric oxide, and induces of IL-10 [10,11]. Interestingly, higher CD39 expression by Tregs could point to its better stability and suppressive activity under inflammatory conditions [12]. However, studies also show that changes in the CD39 and CD73 expression by lymphocytes or other cells can be correlated with distinct clinical profile in a variety of diseases [13,14,15]. These changes in ectonucleotidases levels can impact the purinergic metabolism reflecting in the adenosine availability [16].

Regulatory T cell (Treg) (CD4^+^CD25^high^FOXP3^+^) are known for their lymphocyte-suppressing mechanisms, including the conversion of extracellular ATP into adenosine, and regulatory cytokines expression, such as IL-10, during inflammatory and infectious diseases, such as inflammatory bowel disease and listeriosis [17,18]. However, it is not known whether Treg cells activate the extracellular adenosine pathway during VL. Thus, here we characterized the production of IL-10, as well as the expression of CD39 and CD73 exhibited by Treg cells aiming to better understand the role of this subpopulation in human visceral leishmaniasis pathology. Our results suggest that, although VL led to an increased frequency of Tregs, these cells are not the main source of IL-10. However, the elevated levels of Tregs CD73^+^ may be an attempt to modulate the exacerbation of immune response in VL disease.

## 2. Materials and Methods

### 2.1. Ethical Statement

This study was developed in accordance with the laws and institutional guidelines and in compliance with the ethical standards of the Declaration of Helsinki. Written informed consent was obtained from the volunteers and approved by the Ethics Committee of the Federal University of Paraíba, Brazil (CAAE: 17813013.8.0000.5183).

### 2.2. Study Groups

Samples of the eleven patients with a confirmed diagnosis of VL from the Lauro Wanderley University Hospital of the Federal University of Paraíba were evaluated in absence and presence of SLA stimulation, thus forming the VL MEDIA and VL SLA groups, respectively. The negative endemic control (NEC) group consisted of nine healthy volunteers recruited from endemic areas of the city of João Pessoa, Paraíba, Brazil. 

The inclusion criteria in the VL group were a positive test for the rk39 antigen associated with the pretreatment phase, and the presence of clinical symptomatic disease, characterized by the presence of intermittent fever, hepatosplenomegaly, pancytopenia (erythropenia, thrombocytopenia, and leukopenia), and elevated C-reactive protein (CRP) levels. The NEC group included individuals who were serologically negative for both *L. infantum* antigens and displayed no symptoms. The exclusion criteria were individuals less than 18 years old, with chronic diseases, with physical or mental disabilities, and who personally refused to participate in the study. More details regarding clinical and demographical profile of the volunteers are show in Table 1, Table 2 and Table 3.

### 2.3. Hematological Variables Analysis 

Peripheral blood samples from all volunteers were collected intravenously in tubes without anticoagulant to obtain serum, which was used for the determination of CRP levels, and with anticoagulant (EDTA), which was used for complete blood count. A Hematoclin 2.8 Vet automatic cell hematology analyzer (Bioclin, Belo Horizonte, Brazil) was used to perform blood count, which included red blood cell count, red blood cell width (RDW), hemoglobin, hematocrit (HCT), mean corpuscular volume (CMV), mean corpuscular hemoglobin (MCH), mean corpuscular hemoglobin concentration (CHCM), leukocytes, neutrophils, lymphocytes, monocytes, eosinophils, platelets, mean platelet volume (MPV), and platelet distribution width (PDW). The differential count of leukocytes was determined using blood smears stained with panotic staining and analyzed using an optical microscope. Levels of CRP were determined qualitatively and quantitatively using the latex agglutination method and latex-linked turbidimetry, respectively (EBRAM, Brazil).

### 2.4. Soluble Antigen of L. Infantum (SLA)

*L. infantum* strains (MHOM/BR/1974/PP75) (IOC/L0579) were obtained from the Oswaldo Cruz Foundation (Fiocruz, Brazil). A cryopreserved *Leishmania* was thawed, promastigotes were washed, adjusted to 10^8^ cells/mL in phosphate buffered saline (PBS; Sigma Aldrich, St. Louis, MO, USA; cat.: P3813) followed by three freeze/thaw cycles and a final ultrasonication. Samples were stored at −80 °C until use.

### 2.5. Leukocyte Isolation

Peripheral blood samples from all volunteers were collected in tubes with anticoagulant (heparin) to obtain leukocytes. Leukocytes were freshly isolated from whole blood by red cell lysis (Red Blood Cells Lyses Solution-BD Biosciences, Franklin Lakes, NJ, USA; cat.: 555899). Cells were plated in 96- well U-bottom plates in a concentration of 2.5 × 10^5^ cells per well and cultured for 20 h with either 175 μL of supplemented RPMI medium (Sigma Aldrich; cat.: R8755) (supplemented with 10% of BFS and 1% of Penicillin/Streptomycin) for the NEC and VL groups or 175 μL SLA-containing supplemented RPMI medium for the VL group (at 10 µg/mL final concentration) and used for ex vivo analyses by flow cytometry, according to the manufacturer’s instructions. 

### 2.6. Flow Cytometry Assay

Brefeldin-A (Sigma Aldrich; cat.: B7651; 1 mg/mL) was added during the last 4 h of culture to impair protein secretion, allowing for intracellular cytokine staining. A Human FOXP3 buffer set (BD Pharmigen; cat.: 560098) was used for FOXP3 tagging according to the manufacturer’s instructions. Cells were incubated at 5% CO_2_ at 37 °C for 4 h. Next, to follow the ex vivo protocol, the plaque was centrifugated (8 min, 244× *g*, 4 °C), and the supernatant was removed. Extracellular conjugated antibodies anti-CD4 (APCCy7; clone: RPA-T4), anti-CD25 (PECy7; clone: BC96), anti-CD39 (PE; clone: TU66), and anti-CD73 (APC; clone: AD2) (BD Pharmigen, San Diego, CA, USA/Ebioscience, San Diego, CA, USA) and IgG isotypes control antibodies IgG3-FITC (clone J606, mouse BALB/c IgG_3_, κ, cat. 555578) and IgG1-PE-Cy-7 (clone O4–46, mouse IgG_1_, κ, cat. 561316), were added at the volume suggested by the manufacturer. Cells were incubated for 15 min at 4 °C and later washed with 150 μL of PBS/well. After centrifugation (8 min, 577× *g*, 4 °C) and supernatant removal, extracellular staining was fixed using 100 μL of formaldehyde 4% (Sigma Aldrich; cat.: 252549) diluted in 100 μL of PBS and incubated at room temperature (25 °C) for 20 min. Following centrifugation (8 min, 577× *g*, 4 °C) the supernatant was discarded and samples were washed with 150 μL/well of PBS. Again, the plaque was centrifugated (8 min, 577× *g*, 4 °C) and the supernatant was discarded. To perform intracellular staining, cells were permeabilized with 150 μL of permeabilization buffer (PBS + BSA (0.5%) + Saponin A (Sigma Aldrich; cat.: S7900) (0.5%)) for 10 min at room temperature (25 °C). After centrifugation (8 min, 577× *g*, 4 °C) and removal of the supernatant, intracellular antibodies anti-FOXP3 (Alexa Fluor 488; clone: 259D/C7) and anti-IL-10 (APC; clone: JES3–19F1) (BD Pharmigen, San Diego, CA, USA/Ebioscience, San Diego, CA, USA) were added at a volume suggested by the manufacturer (BD Bioscience, CA, USA). Then, the plate was incubated for 30 min at room temperature (25 °C) and after this 150 μL/well of permeabilization buffer was added. Following centrifugation (8 min, 577× *g*, 4 °C), the supernatant was removed. Finally, 200 μL/well of Wash B (PBS/BSA) was added and the samples were transferred to FACS tubes and stored at 4 °C. At least 50,000 gated events were acquired using FACS CANTO II (BD Biosciences, USA) and analyzed using FlowJo software version 10.4 (BD, Ashland, OR, USA) [19].

### 2.7. Measurement of IL-10 Levels in Leukocyte Culture Supernatant 

Supernatant samples previously obtained after leukocytes incubation in SLA presence or absence and stored at −80 °C were used to measure these levels of IL-10 using ELISA (Human IL-10 ELISA Set kit, BD Bioscience), according to the manufacturer’s recommendations.

### 2.8. Flow Cytometry Data Analysis

The FlowJo software v.10.4 (BD, Ashland, OR, USA) was used to analyze Treg (CD4^+^CD25^high^FOXP3^+^) cells. The establishment of positive and negative populations were performed through the Fluorescence Minus One (FMO), isotype controls, and cells not incubated with any antibodies (negative populations). The analysis strategy used to identify Treg cells included a first gate on lymphocyte area in granularity versus a CD4^+^ plot to identify the CD4^+^ subpopulation (Appendix A). Then, cells with high CD25 expression were identified by means of granularity versus CD25^+^ (Appendix A). Subsequently, the cells that expressed the FOXP3^+^ marker were delimited after the selection of granularity versus FOXP3^+^ cells (Appendix A). Finally, analyses of the respective surface markers and cytokines were performed for each subpopulation.

### 2.9. Statistical Analysis

Statistical analyses were performed using GraphPad Prism Inc. software (Version 6.0) using a unilateral ANOVA and Tukey’s post-hoc test. Significance was determined at *p* < 0.05.

## 3. Results

A total of 20 subjects were evaluated in this study: 11 VL patients and 9 healthy individuals (NEC). Mean age of VL patients and NEC were 43.2 ± 12.8 and 32.6 ± 2.7 years, respectively. Among VL patients, a higher infection rate was observed in men than in women. The NEC group did not present antigens of *L. infantum*. In the VL group, all patients were positive for antibodies to *L. infantum* (SLA) and also had a high serum concentration of CRP (94.6 ± 70.2 mg/L). Finally, a significant reduction in the mean number of erythrocytes (3.4 ± 0.6), leukocytes (2.425 ± 758.7 mm^3^), and platelets (91.700 ± 43.412 mm^3^) was observed in VL patients. Table 1, Table 2 and Table 3 summarizes these findings.

The frequency of Treg cells, which seem to play an important role during the course of VL, was assessed by flow cytometry. The results showed a high frequency of these cells in VL patient groups independent of SLA stimulus compared to the NEC group (Figure 1). The expression of IL-10, a key cytokine in the pathogenesis of VL, was also assessed, and a significant reduction in the production of this cytokine by Treg cells was observed in patients with VL, in the absence of SLA, compared to the NEC group. However, SLA led to the re-establishment of the expression levels of IL-10 by Treg cells in the NEC group (Figure 2A,B). Furthermore, SLA induced a significant increase in IL-10 levels in culture supernatant of VL patients when compared to the NEC group and VL patients without stimulation (Figure 3). 

The expression of ectonucleotidases CD39 and CD73 in Treg cells was evaluated (Figure 4A). Significant changes in CD39 expression and CD39/CD73 coexpression were not observed among the groups (Figure 4B,D). Nevertheless, regarding the frequency of CD73, it was observed that the VL group showed a significant increase in the expression of this ectoenzyme by Treg cells compared to the NEC. Besides that, SLA stimulation tended to maintain the CD73 expression high in VL patients (Figure 4C).

Individuals affected by VL worldwide have similar well-defined clinical and demographic parameters, which guided the selection criteria used here [20]. In agreement with Horrilo et al. (2019), who reported that individuals with VL had an average age of 45 years and were predominantly male, VL patients recruited in the present study were also predominantly male with a mean age of 43.2 years. Adult and economically active people, mainly residing in peripheral regions, comprise the main risk group in relation to VL, which was also observed in the present study [21,22,23].

Additionally, the prevalence of male individuals is unanimous in VL studies mainly because of the association of testosterone and genetic factors with susceptibility to VL [21,24,25,26]. Pancytopenia, fever, and hepatosplenomegaly are considered the main clinical criteria for the classification of VL, and these parameters were observed in all patients evaluated [27]. Elmahallawy et al. (2014) described that the association between serological tests and clinical manifestations is considered a rigorous protocol for the diagnosis of VL, and serological aspects, such as detection of *L. infantum* antigens, were also considered essential for the determination of groups in the present study [28].

## 4. Discussion

Additionally, as expected, individuals evaluated here presented high plasma levels of CRP, a nonspecific marker of the acute phase of inflammatory and infectious processes [29]. In *Leishmania* sp. infections, the opsonization of promastigotes by CRP increases the phagocytosis of these parasites by macrophages. In the intracellular space, parasite survival depends on its inhibitory molecules expression [30]. From an immunological point of view, VL is characterized as a disease marked by the host’s inefficiency in eliminating the parasite, which is mainly determined by the presence of serum IL-10 despite the expressive Th1 profile activation present in the infection [31]. A microenvironment develops, which limits inflammation and permits parasitic replication due to the suppression of IL-12 production and the absence of proliferation and activation of lymphocytes by the action of IL-10, which contributes to the establishment of infection by *L. infantum*, as well as to the chronicity of the disease [3,32,33]. Therefore, because they are recognized as a source of IL-10 in various physiological processes and because of their immunosuppressive performance profile, Treg cells (CD4^+^CD25^high^FOXP3^+^) were evaluated in this study. A significant increase in the frequency of this subpopulation in VL patient groups independently of the SLA stimulus was observed, which may be suggesting an attempt of modulating the immune response established in the disease [6].

The frequency of Treg IL-10+ cells was also assessed. In the absence of antigenic stimulation, a decrease in the frequency of IL-10 produced by Treg cells in VL patients was observed in comparison with NEC group. Considering that the severity of VL disease together with decreased IFN-γ and elevated IL-10 levels, this result can be justified as an attempt by the subpopulation of Treg cells to counterbalance the systemic immunosuppression established by IL-10 in these patients in order to favor the elimination of the parasite through an effective immune response [6,34,35]. In contrast, stimulation of cells with SLA promoted the restoration of IL-10 expression by Treg cells. Rosemblun et al. (2016) reported that peripheral Treg cells are able to respond quickly in cases of antigenic re-exposure, as observed in SLA stimulation performed here, activating their natural suppressive mechanisms, such as the production of IL-10 [36,37].

Given the relevance of IL-10 in the course of VL, culture supernatant levels of this cytokine were also assessed. In the absence of antigenic stimulation, the maintenance of IL-10 levels in patients with VL was observed in comparison with that in the NEC group. In the chronic progressive phase of the disease, the multiplicity of mechanisms present in immunological modulation would justify the maintenance of IL-10 observed at this point [31,36]. This is supported, for example, by the occurrence of inhibitory receptors related to the T cell exhaustion pathway and a mixed production of inflammatory) and regulatory cytokines seeming to act by balancing the levels of mediators produced to guarantee the persistence of the infection despite the host’s efforts to respond. However, with SLA stimulus, culture supernatant levels of IL-10 increase. Thus, these results highlight the IL-10 as the main hallmark of the establishment of VL in the host, originating from different cell sources, including peripheral Treg cells, especially in view of the restoration of its production and its culture supernatant levels in the presence of SLA [36]. 

The metabolism of purines developed by the ectonucleotidases CD39 and CD73 is another known mechanism performed by Treg cells and was analyzed in the present study. No significant changes were observed in CD39 expression and CD39/CD73 coexpression by Treg cells in any of the evaluated groups. However, the VL group showed a significant increase in CD73 expression in these cells compared to controls, only in the absence of SLA stimulation. Thus, the maintenance of CD39 expression suggests that the conversion from ATP and ADP to AMP by these cells remains stable, regardless of the immune status [10,38,39,40]. In addition, the isolated positive regulation of CD73 expression only in the VL group, without stimulation, may indicate that Treg cells do not have the adenosine extracellular pathway as protagonists of their performance in the face of VL infection. Considering that adenosine-mediated inhibitory signaling is used by *L. infantum* to prevent the establishment of an effective host immune response as a survival strategy, because of the marked expression of A2B receptors, the expression profile of ectonucleotidases by Treg cells shown here suggests a modulation in VL mediated by these cells [5,41].

## 5. Conclusions

This study has some limitations, including the low number of patients mainly in the NEC group, which does not necessarily represent the diversity of the Brazilian population. However, even with these limitations, we believe that this work can help the scientific community in understanding the participation of Treg cells in the immunopathology of VL. In this regard, our data demonstrate that Treg cells are not the main source of IL-10 producing cells and that probably there are other sources of this cytokine capable of maintaining the severity of pathology in VL disease. Moreover, the CD73 pathway may be an attempt to modulate by Tregs the exacerbation of immune response seen in this disease. Although our data are preliminary, and despite the need for further studies involving other mechanisms and markers related to this cell subpopulation, our findings are valuable, contributing with the understanding regarding the role of Treg cells in VL disease.

## Figures and Tables

**Figure 1 tropicalmed-08-00018-f001:**
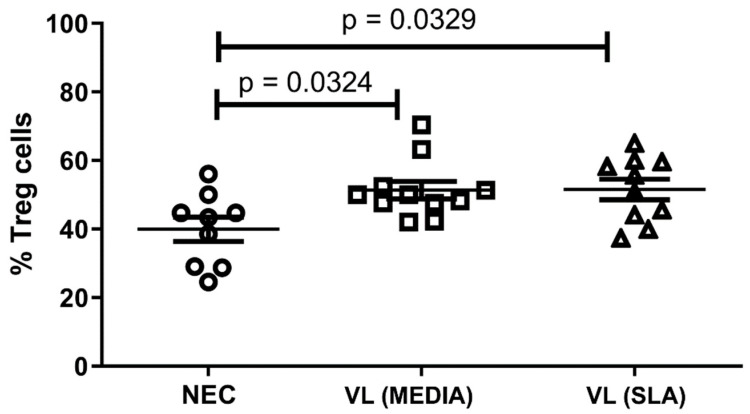
Percentage of Treg cells in peripheral blood of volunteers. Error bars show variability of data regarding the studies groups. ○, □, and ∆ means each volunteer in the negative endemic controls (NEC; n = 9), VL MEDIA (n = 11), and VL SLA (n = 10) groups, respectively. Several groups were compared by unilateral ANOVA and Tukey’s post-hoc test. The lines above bars indicate significant differences (*p* < 0.05) in the compared means of the VL MEDIA, VL SLA, and NEC groups.

**Figure 2 tropicalmed-08-00018-f002:**
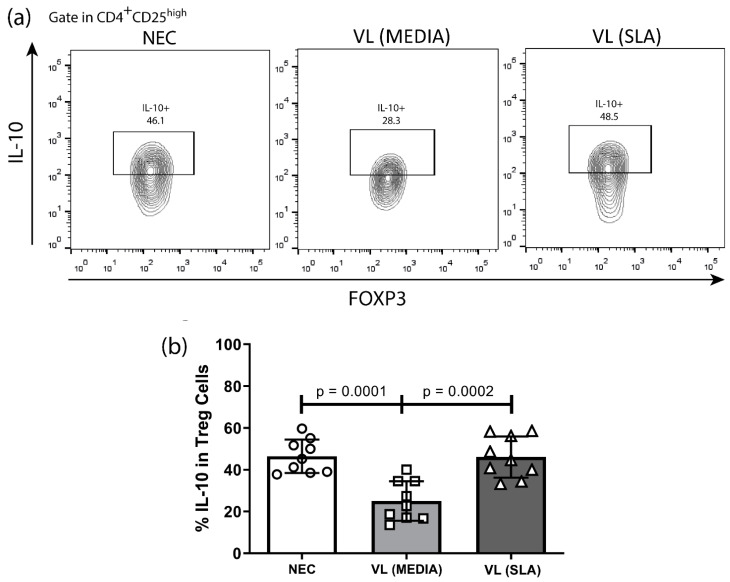
IL-10 expression by Treg Cells in NEC, VL MEDIA and VL SLA groups. (**a**) Representative analyses strategy of IL-10 production by Treg Cells in negative endemic control (NEC; n = 9), VL MEDIA (n = 9), and VL SLA (n = 9) groups. (**b**) Comparative analysis of IL-10 expression by Treg cells between NEC, VL MEDIA, and VL SLA groups. Error bars show variability of data regarding the studies groups. ○, □, and ∆ means each volunteer in the negative endemic controls (NEC; n = 9), VL MEDIA (n = 11), and VL SLA (n = 10) groups, respectively. Several groups were compared by unilateral ANOVA and Tukey’s post-hoc test. The lines above bars indicate significant differences (*p* < 0.05) in the compared means of the VL MEDIA, VL SLA, and NEC groups.

**Figure 3 tropicalmed-08-00018-f003:**
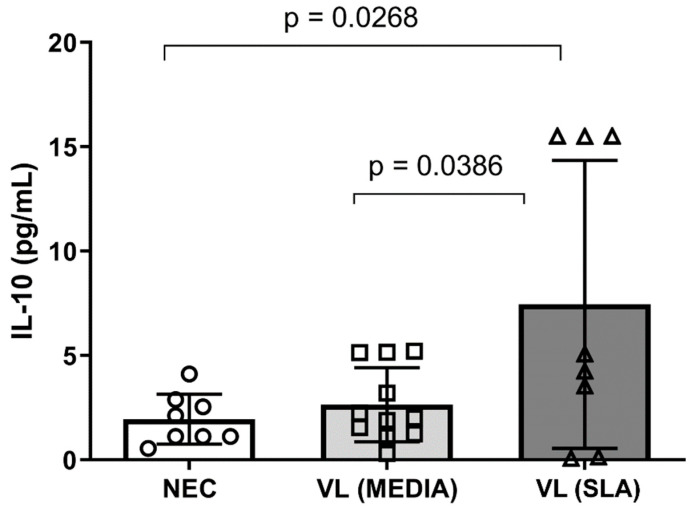
IL-10 level in leukocyte culture supernatant obtained from peripheral blood of volunteers. Error bars show variability of data regarding the studies groups. ○, □, and ∆ means each volunteers in the negative endemic controls (NEC; n = 9), VL MEDIA (n = 11), and VL SLA (n = 10) groups, respectively. Several groups were compared by unilateral ANOVA and Tukey’s post-hoc test. The lines above bars indicate significant differences (*p* < 0.05) in the compared means of the VL MEDIA, VL SLA, and NEC groups.

**Figure 4 tropicalmed-08-00018-f004:**
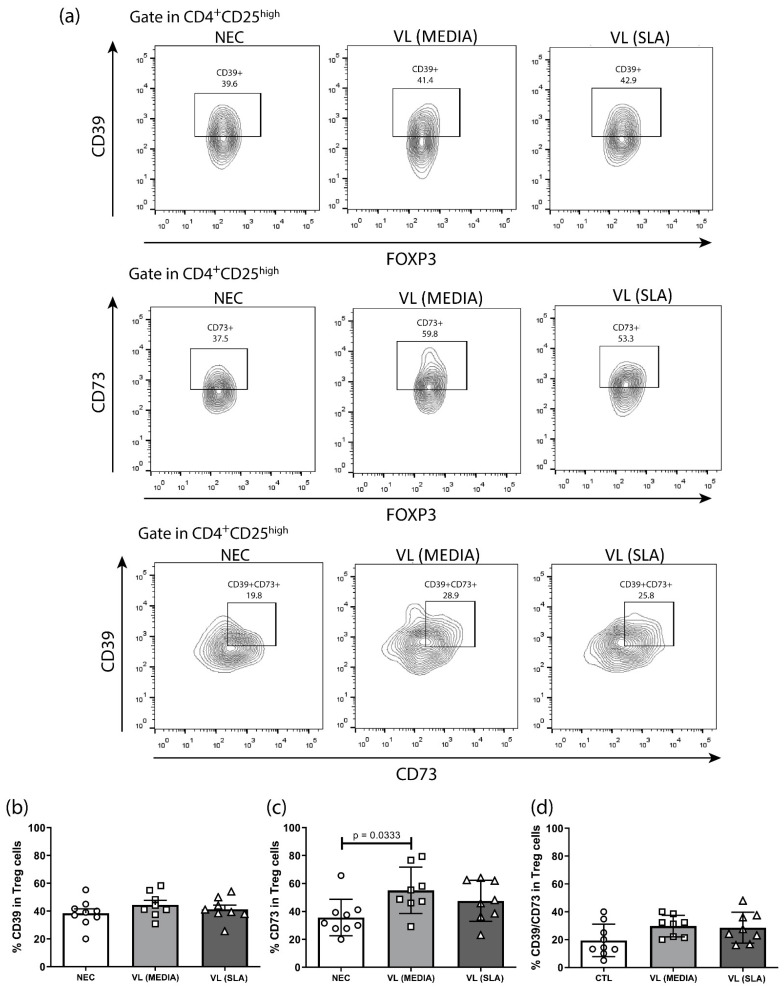
CD39 and CD73 expression by Treg Cells in NEC, VL MEDIA and VL SLA groups. (**a**) Representative analyses strategy of CD39 and CD73 expression by Treg Cells in negative endemic control (NEC; n = 9), VL MEDIA (n = 8), and VL SLA (n = 8) groups and (**b**) Comparative analysis of CD39 expression by Treg cells between NEC, VL MEDIA, and VL SLA groups. (**c**) Comparative analysis of CD73 expression by Treg cells between NEC, VL MEDIA, and VL SLA groups. (**d**) Comparative analysis of CD39 and CD73 coexpression by Treg cells between NEC, VL MEDIA, and VL SLA groups. Error bars show variability of data regarding the studies groups. ○, □, and ∆ means each volunteers in the negative endemic controls (NEC; n = 9), VL MEDIA (n = 11), and VL SLA (n = 10) groups, respectively. Several groups were compared by unilateral ANOVA and Tukey’s post-hoc test. The lines above bars indicate significant differences (*p* < 0.05) in the compared means of the VL MEDIA, VL SLA, and NEC groups.

**Table 1 tropicalmed-08-00018-t001:** Gender and Age of negative endemic control group (NEC group).

#Code	Gender	Age
CTL 1	Female	50
CTL 2	Female	51
CTL 3	Female	22
CLT 4	Female	53
CLT 5	Male	28
CLT 6	Female	25
CLT 7	Female	27
CLT 8	Male	18
CLT 9	Male	20
MEAN	6F:3M	32.6 (±2.7)

F = Female; M = Male.

**Table 2 tropicalmed-08-00018-t002:** Gender, age and symptoms of VL patients (VL MEDIA and VL SLA group).

#Code	Gender	Age	Symptoms
LV01	Female	22	Hepatosplenomegaly, Adnomia, Severe Anemia and Fever
LV02	Male	55	Fever, Asthenia, Anorexia, Weight loss, Sweating, Epistaxis, Jaundice, Cough with expectoration and Hemoptysis
LV03	Male	44	Fever (39 °C), Abdominal pain, Abdominal swelling, Weakness and Dry cough
LV04	Female	29	Tiredness, Dizziness, Fever (38 °C), Abdominal swelling, Abdominal pain, Weight loss and Chills
LV05	Male	35	Abdominal Pain, Headache, Dizziness and Vomiting
LV06	Male	38	Weakness, Dizziness, Hepatosplenomegaly, Fever, Headache, Diarrhea, Anemia and Abdominal pain
LV07	Female	57	Thrombocytopenia, Hepatosplenomegaly, Fever and Anemia
LV08	Male	42	Anorexia, Fever and Asthenia
LV09	Male	37	Fever, Asthenia, Fatigue, Chills, Headache, Nasal Obstruction, Vomiting, Hematuria and Weakness
LV10	Male	53	Fever, Loss of Appetite, Weight Loss and Headache.
LV11	Male	54	Weakness, Headache, Loss of Appetite, Fever, Weight loss, Anorexia, Constipation and Expectoration.
MEAN	3F:8M	43.2 ± 12.8	

F = Female; M = Male.

**Table 3 tropicalmed-08-00018-t003:** Immunohematological aspects of research volunteers.

#Group Code	Anti-*Leishmania* Antibodies (SLA—*L. infantum*)	C-Reactive Protein (CRP) (mg/L)	Red Cells (millions/mm^3^)	Leukocytes (mm^3^)	Platelets (mm^3^)
NEC	0/9	<5	4.5 ± 0.5	6550 ± 1535	243,444 ± 73,428
VL Patients	11/11	94.6 ± 70.2	3.4 ± 0.6 **	2425 ± 758.7 ***	91,700 ± 43,412 ***

Reference values: Red Cells = 4.5–6.1 millions/mm^3^; Leukocytes = 4000–11,000 mm^3^; Platelets = 150,000–450,000 mm^3^; CRP = 5 mg/L. (**) and (***) mean *p* < 0.05 and *p* < 0.001, respectively; Negative Endemic Control (NEC).

## Data Availability

All data supporting the study findings are included in this published article.

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
