# Peer review of "Characterization of Regulatory T Cells in Patients Infected by Leishmania Infantum"

_tropicalmed, 2022, doi:10.3390/tropicalmed8010018_

Round 1

Reviewer 1 Report

In this work, Peixoto and collaborators evaluated the possible correlation of visceral leishmaniasis (VL) with the expression of IL-10, CD39, and CD73 by regulatory T cells. his in patients with VL and negative controls. The authors show that regulatory T cells are not the primary source of IL-10, fundamental for the maintenance of the pathology, opening the possibility that other sources are producing these cytokines.  It is necessary to demonstrate that mechanisms modulate the parasite to evade the immune response and this work makes contributions in that line. This work is revealing, well-written, and with an appropriate methodology.  

Author Response

Point 1: In this work, Peixoto and collaborators evaluated the possible correlation of visceral leishmaniasis (VL) with the expression of IL-10, CD39, and CD73 by regulatory T cells. his in patients with VL and negative controls. The authors show that regulatory T cells are not the primary source of IL-10, fundamental for the maintenance of the pathology, opening the possibility that other sources are producing these cytokines.  It is necessary to demonstrate that mechanisms modulate the parasite to evade the immune response, and this work makes contributions in that line. This work is revealing, well-written, and with an appropriate methodology.  

Response 1: Dear reviewer, we thank you so much for the amazing abstract of the article. We would like to highlight those improvements that were performed in the results and conclusions sections (lines 209, 216, 218 and 352).

Reviewer 2 Report

The manuscript addresses a quite interesting topic but adds only low novelty.  Further there are some important concerns mainly in methods and results.

Major:

The data presented are preliminary. The conclusion that presented results bring a new understanding of the previously established profile for Treg cells in VL disease is exaggerated. The main result of the ms  that Treg are not the main source of IL-10  is obvious.

The group of healthy donors is small. Gender and age dimension of the clinical samples can influence the results and should be statistically analysed.

Clinical and demographic data should be presented for VL MEDIA and VL SLA groups.

The section: 2.5. Leukocyte isolation includes description of flow cytometry methodes. Separate sections for cells isolation and the  flow cytometry should be presented and specificly described.  The used methodology for staining and flow cytometry analysis is not clear. Positive staining and gating strategy should be determined by comparison to an unstained control and a fluorescence minus one (FMO) control.

Statistical analysis should be described as a separate section.

Minor:

Introduction. The role of CD39 and CD73 in  immune cells with the pathophysiological context (“immunological switches” ) should be described.

Author Response

Point 1: The data presented are preliminary. The conclusion that presented results bring a new understanding of the previously established profile for Treg cells in VL disease is exaggerated. The main result of the ms that Treg are not the main source of IL-10 is obvious.

Response 1: Dear reviewer, we thank you for the notes. Aiming to avoid misunderstanding regarding our results, we performed little modifications in the introduction (lines 62) and conclusions sections (lines 352 and 361). We hope that these changes can be in compliance with your observations.

Point 2: The group of healthy donors is small. Gender and age dimension of the clinical samples can influence the results and should be statistically analysed.

Response 2: Dear reviewer, we are grateful for the appropriate observation and agree with the comments. We understand that this study has some limitations, including the low number of patients, mainly in the Healthy group. However, even with these limitations, the present study showed relevant and statistical results regarding the Regulatory T cells in human visceral leishmaniasis (VL) that could help the scientific community understand the participation of these cells in the immunopathology of VL. But, thinking about your comment, we decided to add a little observation about it the conclusion section (page 10, lines 352-356).  Regarding the statistical analyses, due to the low number of volunteers enrolled in this study, we did not perform them. However, we thank you for the suggestion and intend to do this kind of analysis in future studies.

Point 3: Clinical and demographic data should be presented for VL MEDIA and VL SLA groups.

Response 3: Dear reviewer, thank you for your comment! Aiming to detail the clinical and demographic data we created 3 tables that encompasses these data (line 209, 216 and 218). However, the volunteers that were enrolled in VL MEDIA and VL SLA groups are the same. The difference between both is the stimulation profile according to subsection “2.2 Study Groups” (line 74).

Point 4: The section: 2.5. Leukocyte isolation includes description of flow cytometry methodes. Separate sections for cells isolation and the flow cytometry should be presented and specificly described.  The used methodology for staining and flow cytometry analysis is not clear. Positive staining and gating strategy should be determined by comparison to an unstained control and a fluorescence minus one (FMO) control.

Response 4: Dear reviewer, we are grateful for the appropriate observation. We separated the cell isolation and flow cytometry methodology. In this regard, we created a new section called “2.6 Flow Cytometry Assay” while Leukocyte isolation remained in this same number subsection. Moreover, the subsection “Cell surface and single-cell cytoplasmic cytokine staining” was addicted in the subsection “2.6 Flow Cytometry Assay” (Line 120).  Also, an improvement in the flow cytometry methodology was performed, aiming to clarify the information better. Finally, we added the FMO in S1 figure that supports our positive staining and gating strategy (Line 194). 

Point 5: Statistical analysis should be described as a separate section.

Response 5: Dear reviewer, thank you for your suggestion. In fact, the flow cytometry data analysis and statistical analysis were in the same subsection. In this sense, we divided that in two subsections named: “2.8. Flow cytometry data analysis” and “2.9. Statistical analysis” (lines 173 and 187).   

Point 6: Introduction. The role of CD39 and CD73 in immune cells with the pathophysiological context (“immunological switches”) should be described.

Response 6: Dear reviewer, we are grateful for the appropriate comments, and we understand that this suggestion can enrich this paper. Thus, we added five references about the involvement of the ectonucleotidases expression in some pathologies in the introduction sections (lines 50-55).

Round 2

Reviewer 2 Report

I'm satisfied with the improvement.